# Polymer Supercritical CO_2_ Foaming under Peculiar Conditions: Laser and Ultrasound Implementation

**DOI:** 10.3390/polym15081968

**Published:** 2023-04-21

**Authors:** Jennifer Andrea Villamil Jiménez, Margaux Haurat, Rayan Berriche, Fabien Baillon, Martial Sauceau, Mattéo Chaussat, Jean-Marc Tallon, Andrzej Kusiak, Michel Dumon

**Affiliations:** 1Centre RAPSODEE, UMR CNRS 5302, IMT Mines Albi, Université de Toulouse, F-81013 Albi, France; jennifer.villamil_jimenez@mines-albi.fr (J.A.V.J.);; 2Laboratoire de Chimie des Polymères Organiques (LCPO), UMR 5629, Bordeaux INP/ENSCBP, University Bordeaux, CNRS, F-33607 Pessac, France; margaux.haurat@u-bordeaux.fr (M.H.); rberriche@bordeaux-inp.fr (R.B.); 3Département Science et Génie des Matériaux -SGM, IUT Institut Universitaire de Technologie, Université de Bordeaux, F-33170 Gradignan, France; 4I2M Laboratory, UMR 5295, Bordeaux INP, University Bordeaux, CNRS, F-33400 Talence, France; matteo.chaussat@etu.u-bordeaux.fr (M.C.); andrzej.kusiak@u-bordeaux.fr (A.K.)

**Keywords:** polymer, PMMA, foaming, sc-CO_2_, laser, ultrasound

## Abstract

The two-step batch foaming process of solid-state assisted by supercritical CO_2_ is a versatile technique for the foaming of polymers. In this work, it was assisted by an out-of-autoclave technology: either using lasers or ultrasound (US). Laser-aided foaming was only tested in the preliminary experiments; most of the work involved US. Foaming was carried out on bulk thick samples (PMMA). The effect of ultrasound on the cellular morphology was a function of the foaming temperature. Thanks to US, cell size was slightly decreased, cell density was increased, and interestingly, thermal conductivity was shown to decrease. The effect on the porosity was more remarkable at high temperatures. Both techniques provided micro porosity. This first investigation of these two potential methods for the assistance of supercritical CO_2_ batch foaming opens the door to new investigations. The different properties of the ultrasound method and its effects will be studied in an upcoming publication.

## 1. Introduction

The design of porous polymers has become an area of great interest and the methods of fabrication of these materials are numerous. In general, foaming involves an expansion of the material after the dissolution of a blowing agent that can be chemical or physical. Between the physical blowing agents, supercritical fluids have emerged as the best option, with sc-CO_2_ and sc-N_2_ being the most employed [1,2]. CO_2_ is a well-known, cheap, non-toxic, non-flammable, inert and highly available pure gas, and its critical conditions can be easily reached (T_c_ = 31 °C, P_c_ = 7.38 MPa). Sc-CO_2_ has been largely used as a blowing agent for the different foaming process, i.e., discontinuous (batch), semi-discontinuous (injection) and, continuous (extrusion), and with a wide range of polymers such as polymethyl methacrylate (PMMA), polycarbonate, polyethylene terephthalate (PET), polystyrene (PS), glycol-modified PET, polyvinyl chloride (PVC), polypropylene (PP), polyurethane (PU), polyimide and polycaprolactone [1,2]. Nevertheless, (a) generating a great cellular density (i.e., ≥10^15^ cells/cm^3^), (b) stabilising growing bubbles at an early stage of the process and, (c) obtaining low densities are still challenges. Creating a large number of small bubbles simultaneously at the beginning of the foaming is a difficult task. The properties and morphologies of the foams can be variated by varying either the material design, employed blowing agent, operating conditions (saturation temperature, pressure and time), depressurisation rate (dP/dt), special coolers or mixers, etc. Moreover, new techniques to manage the cellular morphology of the foams were developed including foaming in different stages, enhancing crystallinity or the polymer matrix, and the use of external “actions” such as laser irradiation (LS) and ultrasonic waves (US).

The present work concerns the assistance of (a) LS and (b) US. Our preliminary work on LS-aided scCO_2_ foaming is new and LS-aided foaming is nearly absent from our literature review. Since we will deal mainly with US, the foaming assisted by laser will be not presented in this introduction.

Few papers studying foaming process aided by additional techniques, such as ultrasonic waves, are currently available. Ultrasonic irradiation (US) is mainly used in industry for filtration, freezing, drying, separation, extraction, cleaning, mixing, emulsion sterilisation, diagnosis, defect detection, etc. [3,4,5]. US is divided into low frequency (16 to 100 kHz), high frequency (>100 kHz) and very high frequency (>1 MHz). Most of the US cleaning baths available in laboratories operate with frequencies between 35 to 45 kHz and, have powers from 100 to 500 W. The mechanisms of action of ultrasound on materials are divided into thermal and non-thermal. Thermal effects occur when the acoustic energy is absorbed and transformed into heat. Such an effect depends on the absorption, frequency and dissipation of the ultrasound energy. Ultrasound can act by (i) cavitation (formation of vapour bubbles of a flowing liquid in a region where the pressure of a liquid falls below the vapour pressure) and by (ii) mass transfer enhancement. US has also been combined with other technologies such as microwaves, supercritical CO_2_ extraction, high pressure processing and enzymatic extraction [6]. 

The effect of ultrasound can also depend on the state of the polymer. In the liquid state (polymer melt), apparent viscosity reduction, molar mass distribution decrease, crosslinking, increase in the motion of liquid molecular chains and plastic welding can be observed when using ultrasound [7,8]. In the work of Yang. et al. [8], an increase in the melt strength of polypropylene was observed after the application of 300 W of ultrasonic waves that induced chain scission and recombination reactions at the die where an ultrasonic probe was set. In regard to foaming, Yang et al. [9] used a chemical foaming agent (Azodicarbonamide) to produce a cellular polymer. When US and talc were introduced into the high melt strength polypropylene (HMSPP) melt, the HMSPP foam had a larger cell density and smaller size as well as a more uniform distribution. Regarding thermal properties, compared to the HMSPP foam without ultrasonic oscillation, the minimum thermal conductivity of the HMSPP foam with US reached 0.055 W/m/K. The ultrasound-assisted chemical extrusion was said to be a promising technology to continuously and rapidly fabricate better polymer foams.

In the solid state, a reduction of the intrinsic viscosity or the molecular weight when increasing the frequency of applied ultrasound to the two water-soluble polymers carboxymethyl cellulose (CMC) and polyvinyl alcohol (PVA) was observed, evincing a degradation of the polymers [10]. In the work of Price et al. [11], powders of polyethylene (PE), PP, PVC and PMMA were subjected to irradiation with high-intensity ultrasound while suspended in water; changes in the particle sizes and the surface morphology were obtained, and these changes could be correlated with the physical properties of the evaluated materials. Regarding the foaming process, the literature contains rather few examples of US-aided microcellular foaming processing, usually by heating simultaneously the gas-saturated polymer and employing ultrasound waves in a solid state. Adam et al. [12] foamed PS using a temperature-induced batch foaming assisted by sc-CO_2_. The nucleation was triggered by the elevation of the temperature of the super-saturated sample, in this case, the ultrasound were applied during this stage. At a foaming temperature of 50 °C, the cell density of the foams was increased, evincing that the use of ultrasound reduced the nucleation barrier. The cell size ranged from 0.5 to 3.5 µm, with or without US applied at 60 or 70 °C. The smallest cell size (0.3–2.4 µm) was obtained at 50 °C with the aid of ultrasound.

Gandhi et al. [13] found that there is a critical effective distance from the ultrasonic transducer where the cavitation is maximal and beyond which, the intensity of it decreases drastically. It was found that only when foaming (one-step) acrylonitrile–butadiene-styrene (ABS) at the critical effective distance, a remarkable increase in the cellular density could be observed. The expansion ratio and cell morphology were also found to be significantly affected by the relative placement of gas-saturated polymers with respect to the transducer in the sonication medium. In another study, Gandhi et al. [14] also showed that longer periods of ultrasound exposure developed foams with smaller cell sizes. The ultrasonic frequency was also found to significantly influence the morphology. Low-frequency sonication resulted in foams with a uniform cell distribution, whereas high frequency sonication developed a bimodal microcellular type of microstructure.

Wang et al. [15] foamed polylactic acid (PLA) using sc-CO_2_ as the blowing agent in a temperature-induced batch process (two-step foaming). Ultrasound was applied during 60 s. It was noticed that before the ultrasound application, most pores were closed. After the ultrasound exposure, the pores became mostly open. This shows that inter-pore connectivity of the foams can be substantially enhanced by applying ultrasound treatment.

This work studied the foaming of amorphous PMMA [16]. A solid-state, two-step sc-CO_2_ batch foaming process was employed, and the assistance of laser irradiation (LS) and ultrasonic waves (US) on the cellular morphology of the foamed samples was investigated. US was especially tested to initiate more nuclei, increase the nuclei density, and break growing bubbles towards micro and nano foams. Measurements of thermal conductivity (λ W/m/K) were also carried out in order to examine a possible effect of US. 

## 2. Materials and Methods

An extrusion grade PMMA (Altuglas ^®^ V825T) produced by Arkema ^®^ (Serquigny, France) was employed. The characteristics of PMMA are: M_n_ = 43,000 g/mol, M_w_ = 83,000 g/mol, T_g_ = 115 °C, amorphous transparent polymer, and bulk density 1.19 g/cm^3^. 

### 2.1. scCO_2_ Saturation Stage

All samples were first impregnated with CO_2_ for 24 h at 40 °C and 10 MPa using two supercritical CO_2_ units [(i) in LCPO Bordeaux, 0.5 L autoclave from Separex, (Champigneulles, France); (ii) in RAPSODEE Albi, extraction unit SF 2 × 0.5 L 4158 pilot, developed by Separex (Champigneulles, France)]. Injected bars 80 × 10 × 12 mm in size were used for LS and conventional US. For localised US, they were cut into smaller samples 10 × 10 × 5 mm in size from which two samples were saturated at the same time in the autoclave to compare the foaming with and without localised US. To avoid foaming during the release of the gas and to ensure that the samples are in a transparent state after the saturation stage, the depressurisation profile similar to the one shown in Figure 1 was applied.

### 2.2. Laser Assisted Foaming

A classical CO_2_ laser for cutting or engraving materials (Trotec SPEEDY 300, FabLab IUT Bordeaux, Bordeaux, France), with a power capacity of 20 to 120 W and scanning capacity of 610 × 305 mm, was used to heat the sample. For this, the sample was scanned by the laser beam. The samples were fixed into the laser equipment after their saturation with CO_2_. 

The engraving mode was chosen to initiate the foaming of the PMMA samples line by line (Figure 2).

Scanning speed (S) is expressed as % of maximum speed; here, S = 10 or 50%. Laser power (P) is expressed as % of maximum power; here, P= 20 or 90%, i.e., respectively 24 W and 108 W. Pulsation per inch (PPI = 1000) and ‘0′ offset in z (distance between material and laser) are constant values.

### 2.3. Ultrasound-Assisted Foaming

#### 2.3.1. Conventional Ultrasound Bath

A conventional ultrasound water bath was used in “normal” mode (no pulsation, no sweep, degas modes; Elmasonic S 30 H (Elma Electronic, Strasbourg, France) dimensions of tank: 240 × 137 × 100 mm) with a working frequency of 37 kHz, an ultrasonic power of 320 W (at peak frequency), and an effective ultrasonic power of 80 W. The water temperature was set from 30 up to 100 °C. Three temperatures were chosen; the bath was first regulated for 15 min at 50 °C, 80 °C and 100 °C before immersing the CO_2_-saturated samples for 90 s (maintaining them inside the water bath—below the water upper level—by a heavy glass beaker).

#### 2.3.2. Localised Ultrasound

In Figure 3, the used setup is shown. Inside a distilled water bath, two beakers were placed and filled with distilled water. An ultrasound probe was placed in one of the beakers, and the distance between the probe and the bottom of the beaker was kept at 10 mm for all the tests. A Vibracell probe model 72441 by Bioblock Scientific (Fisher Bioblock Scientific, Aalst, Belgium) was used. This probe has a maximum power capacity of 600 W and a frequency of 20 kHz. An electronic power of 300 W was applied. After the saturation step, one sample was plunged into the beaker with just water (without US). The other sample was plunged into the beaker with the ultrasound probe (with US), and the samples were always placed under the probe. Three bath temperatures (T_foaming_) were tested: 50 °C, 73 °C and 80 °C. The with and without ultrasound samples were plunged into the beaker for 30 s.

### 2.4. Density and Porosity 

Water pycnometry (buoyancy method) was employed to measure the density of the samples (ρ) and the total porosity ($\varepsilon_{T}$) and the expansion ratio (E) of the foams were calculated as follows:(1)$$\varepsilon_{T}=1-\frac{\rho_{f}}{\rho_{s}}\;\mathrm{and}\;E=\frac{\rho_{s}}{\rho_{f}}$$

$\rho_{f}$: foamed sample density (g/mL) 

$\rho_{s}$: solid sample density (1.19 g/mL)

### 2.5. Cellular Morphology

For the study of the cellular morphology of the produced foams, scanning electron microscopy (SEM) was employed (model HITACHI S-3000N). For the preparation of the samples, foams were frozen in liquid nitrogen and fractured to assure that the microstructure remained intact. The surfaces were coated with gold using a sputter coater (model EMSCOPE SC 500), in argon atmosphere. 

Cell size distributions were estimated by measuring the maximum Feret diameter of the observable cells on the SEM microphotographs using ImageJ software.

Cell density ($N_{0}$, expressed in cells/cm^3^) was estimated using Equation (2).
(2)$$N_{0}=\left(\frac{6\times 10^{21}}{{\pi \Phi }^{3}}\right)\left(\frac{\varepsilon_{T}}{1-\varepsilon_{T}}\right)$$
where Φ is the average mean diameter of cells, expressed in nm, and $\varepsilon_{T}$ is the porosity (calculated from Equation (1)). It is a rough estimation, since cell distributions are sometimes large. 

### 2.6. Thermal Conductivity 

The thermal conductivity of the materials was measured by the thermal plane source method using a TPS2500S apparatus. The thermal plane source consists of monitoring the temperature response due to heating by a disk source over a time t. The 7577 sensor with heating area of 4 mm in diameter was used. This sensor was chosen in order to ensure that the possible probing depth was at least twice that of the sensor diameter and fulfilled the assumption that heat does not diffuse to the sample boundaries during measurement time. For each measurement, two identical samples of the material were places in contact with the sensor and pressed without deforming them in order to ensure a good contact between the sensor and the sample. When supplied with a constant electrical current, the nickel spiral of the sensor generates Joule heating and at the same time permits the monitoring of the average temperature ΔT of the heated area through the evolution of its electrical resistance.

In this configuration, the theory gives the time-dependent temperature increase as:(3)$$\Delta T\left(\tau \right)=\frac{Q}{\pi^{3/2}\;r\;k}D\left(\tau \right)$$
where Q is the total heat flux generated by the sensor, r is the radius of the sensor, k is the thermal conductivity of the sample and $D\left(\tau \right)$ is a geometry-related function according to dimensionless time $\tau \;=\sqrt{t/\theta }$. In the expression of τ, t is the time and θ is the characteristic time defined as: $\theta =r^{2}/a$ where a is the thermal diffusivity of the sample. Denoting the total measurement time $t_{m}$, a ratio of total measurement time to thermal characteristic time $\mathrm{TCT}=t_{m}/\theta$ can be defined.

In order to ensure reliable results, two crucial parameters were chosen: the heating power Q and the measurement time $t_{m}$. These two setting parameters are generally fixed through a trial and error approach. The heating power Q was chosen to obtain a temperature increase not exceeding 5 K over the measurement time in order to avoid any nonlinear behaviour of the sample material. On the other hand, the measurement time was adjusted to allow the heat to diffuse sufficiently into the sample but without achieving its outer boundaries. In order to validate the selected Q and $t_{m}$, three criteria were evaluated from the realized measurements: the probed depth δ which must not exceed the sample geometry, the ratio of total measurement time to thermal characteristic time TCT which is recommended to be in the interval between 0.33 and 1, and finally the mean value of residuals $\sigma_{m}$ corresponding to the difference between measured and calculated (Equation (3)) temperature and which is expected to be lower than 10^−3^ K.

## 3. Results and Discussion

### 3.1. Laser-Assisted Foaming

As soon as a line is scanned, PMMA whitened, evincing a foaming process. Figure 4 shows the sample surface of pure PMMA after the whole predefined scans, after the complete scanning lines. Nevertheless, these whitened zones peeled off to make foamed chips.

As described in the experimental section, the procedure followed in this laser-assisted foaming (with this specific equipment) only affected the surface. Therefore, an interesting perk of this technique is that the foaming localization can be controlled since there was no expansion outside of the laser scan. 

The surface morphology observed by SEM (Figure 5) revealed rather homogeneous pores, especially at the highest power and speed (90%, 100%). The fastest scanning speeds can probably freeze the pore structure (rapid crossing below T_g_). In these conditions, the diameters were in the micron range. Additionally, a pore elongation seemed to appear in some areas; its origin could be due to the scanning mode of the laser. 

If the power was increased and the speed lowered, the PMMA started to degrade. Nevertheless, this type of foaming assistance appears to be a useful method for local foaming to induce micro porosity. The literature contains very few cases and also reveals that foaming only occurs on the surface [17,18]. 

As a conclusion, the main result of this preliminary study of laser-induced gas saturated-polymer foaming is the possibility of local foaming at the micron level onto a surface. This is, to our knowledge, the first example of such result. 

### 3.2. Ultrasound-Assisted Foaming

#### 3.2.1. Conventional Ultrasound Bath

Figure 6 shows examples of the obtained foams at 50 °C without or with US. In Figure 7, porosity ($\varepsilon_{T})\;$ classically increased with temperature, but there was also a supplementary effect of the ultrasound, that was only appreciable at low temperatures (50 °C).

It was assumed that the temperature of the bath was not affected by the use of ultrasound during the time of the experiments. In Figure 8, it is observed that the higher the temperature, the higher the expansion ratio and the lower the density; this behaviour is again well known for this kind of process [1]. However, when exposing the samples to the ultrasound at the same temperature, the density was further decreased; this effect was overcome by temperatures above 80 °C. 

In Figure 9, the cell density and average cell size are shown. It can be noted that, in general, when increasing the temperature, the cell density increased for both with and without ultrasound samples. Regarding the effect of the ultrasound at the same temperature, it can be noted that only for a temperature of 100 °C the cell sizes decreased very slightly; in the other cases, the cell sizes in the same range (even very slightly increased). At each temperature, the effect of US for increasing the cell density was probably the most interesting, although the behaviour is not very significant.

Figure 10 shows SEM micrographs of the porous structures. It can be observed that less homogeneous structures were obtained with increasing temperature; nevertheless, when exposing the samples to ultrasound at the same temperature, more homogeneous structures were visually observed. In melt foaming, the two previous effects (on N_o_ and Φ) were noted in extruded PP foamed by a chemical blowing agent in the presence of talc [9]. In solid-state foaming, these US effects were slight as noted by one previous work on PS foams [12]. Our work on neat PMMA foams is thus consistent with these two studies.

As a first preliminary conclusion, US seems to act as a foam modifier and homogenizer, and acted towards increasing the nuclei densities and lowering the foam density.

#### 3.2.2. Localised Ultrasound

Figure 11 shows the initial and foamed PMMA samples using localised ultrasound at a foaming temperature of 80 °C. Figure 12 shows the density of the solid PMMA and the produced foams; it can be observed that as the temperature increased, the density was reduced for both ultrasound and non-ultrasound foams. Regarding the effect of the localized ultrasound, there was a reduction in the density which was more noticeable when increasing the foaming temperature. The effect of US was two-fold: on the one hand, it enhanced cavitation (bubble generation on gas and liquids, thus nucleation) and on the other hand, it breaks structures mechanically (generates cracks, break fibres, etc.). Increasing power or time exposition may be thus detrimental. This is the reason for choosing a short exposition time (30 s) to detect a potential effect on nucleation only, knowing that long times will favour growth and coalescence. In this way, the effect of localized US in the solid state is better using short times (30 s was chosen) and low temperatures (here, 50 °C) and it seems logical that US effects become negligible at high temperatures (e.g., 80 °C).

In Figure 13, the total porosity of the different obtained samples can be observed. Increasing the temperature led to a classical increase in the porosity.

Comparing the nature of the ultrasound (conventional vs. localized), there seemed to be some benefit to applying localized US in our conditions. Such a result is still being studied. At first glance at the SEM micrographs (Figure 14), localized US seemed to induce more small cells and fewer large cells, with a reduction in the average pore size at 50 °C (and a narrower distribution). As for conventional US, the effects of localized US became attenuated at 80 °C. 

Thus, comparing conventional and localised US is not straight forward. Other important factors are the (i) duration of foaming under US and (ii) size of the sample. US may have a benefit to be revealed by changing the power, temperature or localization (maximum cavitation) as stated by reference [13]. 

Further work (distributions of cell sizes, effect of US power, time of exposure, etc.) is under progress to understand the phenomena. This work combines PMMA with a tri-block copolymer (BCP) named MAM (polymethylmethacrylate-co-polybutylacrylate-co-polymethylmethacrylate) [16] for further enhancing nucleation. 

Figure 15 shows the (a) frequency and (b) cumulated frequency of the cell sizes of the different samples. The first observation to be made is that for a temperature of 80 °C, there was no influence from the use of ultrasound on the mean cell diameter (3 µm). The homogeneity of the cell structure was not affected, and the median absolute deviation (MAD) corresponded to 0.8 µm for both distributions. When decreasing the temperature to 50 °C, a decrease in the cell sizes were observed. Exposing the samples to ultrasound resulted in a slight decrease in the mean cell size from 2.5 µm to 2.1 µm. Regarding the MAD, a more homogeneous structure was obtained when applying the ultrasound, decreasing from 0.7 µm to 0.5 µm. These results suggest that localised ultrasound only has effects on cell size and cell size distribution at lower temperatures.

Figure 16 shows the cell density of the samples. It can be observed that there was no difference between the cellular densities between the samples produced at 50 °C and 80 °C. Therefore, regarding the effect of ultrasound, an increase of the cell density was noticed for both temperatures. There are two possible reasons: firstly, the ultrasound reduced the required energy to nucleate a cell or the ultrasound provided extra energy to the system, which allowed the cells to nucleate faster.

### 3.3. Comparison of Thermal Properties after Conventional Ultrasound-Assisted Foaming

#### 3.3.1. Setting the Thermal Characterisation Parameters

Prior to thermal characterization, a comprehensive assessment for the proper selection of Q and $t_{m}$ was realized on a foamed PMMA sample. The sample used for this assessment was foamed according to the following protocol: saturation with sc-CO_2_ for 24 h under 10 MPa of pressure at 40 °C, then foaming in an 80 °C water bath. Measurements sweeping different values of Q and $t_{m}$ were realized. As the materials were expected to be poor thermal conductors, the range of Q was between 3 and 14 mW. With the dimensions of the samples being small (less than 12 mm), the explored measurement time ranged between 4 and 40 s. Figure 17a–e presents the maps of the obtained values of the listed criteria. Firstly, one can observe in Figure 17a that the temperature increase was lower than 2 K for heating powers below 5 mW even when the measurement time was 40 s. This signifies that the applied power should be more than 5 mW. Next, from Figure 17b, one can conclude that a satisfying probing depth, at least on the order of the sensor size, can be obtained using measurement times of 20 or 40 s.

Figure 17c shows the TCT ratio, which appears satisfying (0.4–06) for measurement durations of 20 or 40 s. However, when one checks the mean of residuals (Figure 13d), it exceeded the recommended value below 10^−3^ K for a measurement time of 40 s. It also exceeded the recommended value for the lower measurement duration (20 s) when the heating power was set to 14 mW. These criteria indicate that the most adapted heating power is 6 mW or slightly more and a measurement time of 20 s. It appears from Figure 17e that the thermal conductivity for measurement times lower than 10 s seemed to be overestimated. In turn, for the 40 s measurement time, the values of k were the lowest found, but the residuals exceeded the recommended value and these results must be discarded.

For the measurements on all the produced samples, the sensor power was set to 6 mW in order to avoid any overheating and the measurement time was 20 s. The measurements were repeated 10 times for each characterized sample and the mean values of thermal conductivity were determined.

#### 3.3.2. Thermal Conductivity of Materials Obtained by Foaming without and with Conventional Ultrasonic Assistance 

Concerning the thermal conductivity of the obtained foams when using the conventional US bath, not surprisingly, it first followed the trend of the evolution of the bulk density previously observed; namely, it decreased with decreasing density. 

Next, an influence of ultrasonic assistance was observed, especially at the lower temperatures used. This effect disappeared at 100 °C (foaming was governed only by T). The US assistance led to a decrease in thermal conductivity of around 10% for foaming at 80 °C and 15% when the foaming bath was at 50 °C (Figure 18). Thus, the benefits of applying US were obvious after measuring thermal properties; the reasons are not yet clear and are under investigation on samples fabricated by localized US. It appeared that even if bulk densities lie in the same range of values, thermal conductivity may be affected by US (change in the distribution of cell size, open cell content, connectivity, wall thickness, etc.).

## 4. Conclusions

Two peculiar assistance methods were tested for polymer gas foaming after saturation of samples by scCO_2_: heating by laser scans (LS-aided foaming) or thermal foaming in an ultrasound water bath (US-aided foaming). The foaming process itself is traditionally named a two-step, solid-state batch scCO_2_ process, meaning that, foaming occurs in a second heating step. Here, such an assistance/aid was achieved either with the aid of LS or US. Both aids are easy-to-apply technologies and revealed advantages and difficulties. 

LS offers the possibility of local foaming at the micron level onto a surface by a normal laser engraving mode (“line by line”), while the bulk major part remains unfoamed. LS is a surface foaming aid. 

US is a temperature bulk foaming aid (in a solid-state, two-step foaming process). The US aid was revealed to be an efficient technique to improve foam homogeneity, to decrease the bulk density and cell size and to increase the cell density. One difficulty comes from the control of the exposure time, power and localisation of the applied US. This opens a world of possibilities to master the cellular morphology of the foams just by varying the ultrasound parameters. Coupling power, time and temperature requires a balance in order to favour cavitation and nucleation, avoiding “mechanical” damage of the foam structures. 

The perks of using ultrasound were confirmed after measuring thermal properties, which showed a decrease in the foam thermal conductivity. 

## Figures and Tables

**Figure 1 polymers-15-01968-f001:**
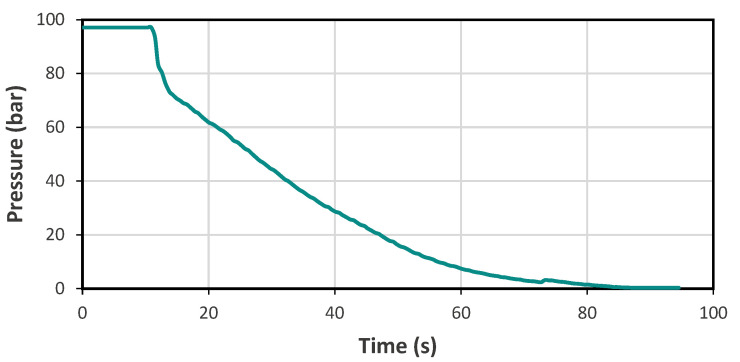
Depressurisation profile.

**Figure 2 polymers-15-01968-f002:**
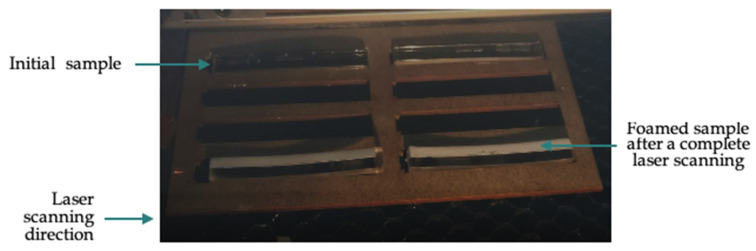
Laser foaming experimental setup with inserted samples.

**Figure 3 polymers-15-01968-f003:**
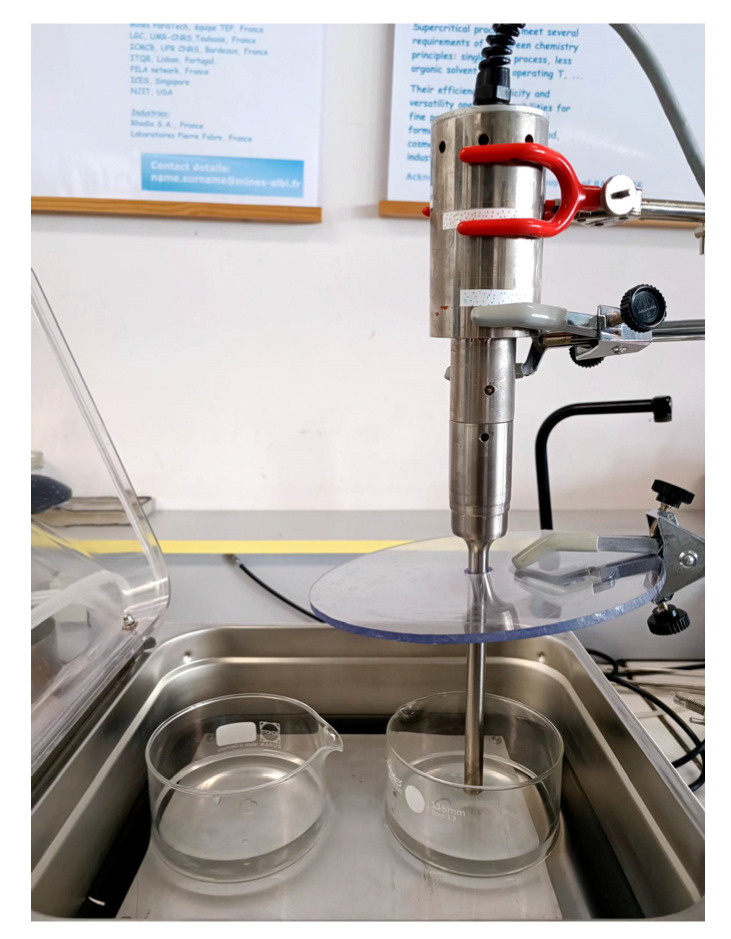
Setup used for foaming process assisted by localised ultrasound.

**Figure 4 polymers-15-01968-f004:**
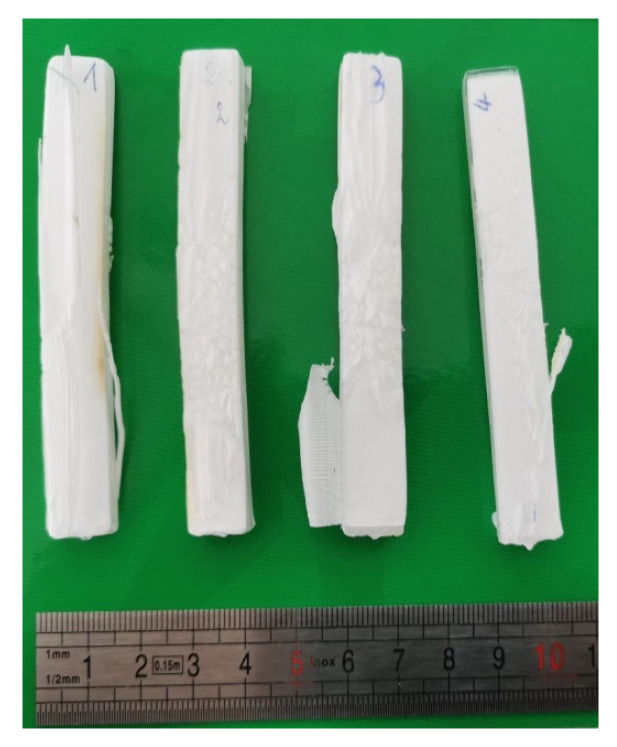
Foamed PMMA samples after laser heating scans.

**Figure 5 polymers-15-01968-f005:**
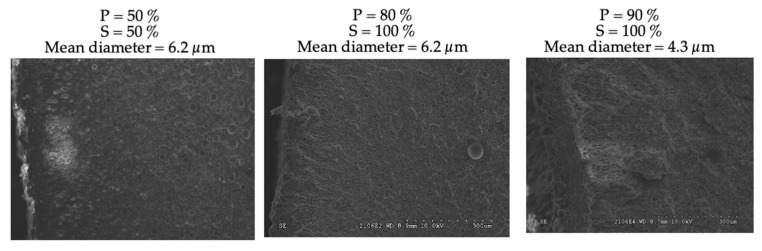
SEM of foamed PMMA surface morphologies after laser heating at different scanning speeds (S) and power (P).

**Figure 6 polymers-15-01968-f006:**
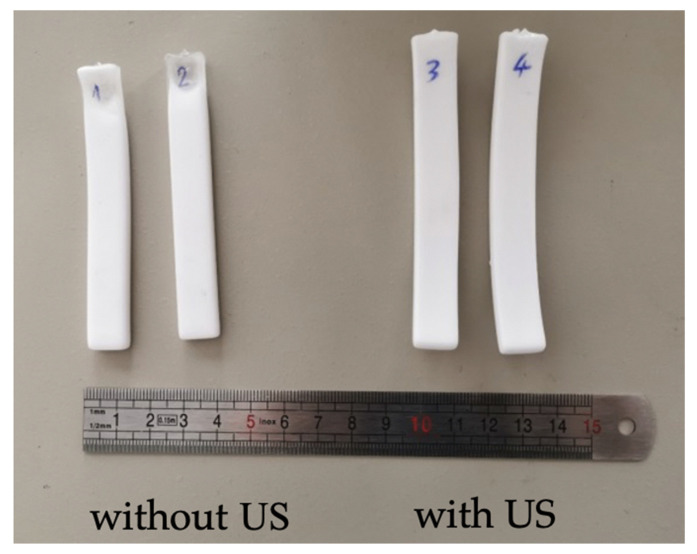
PMMA samples foamed without and with conventional ultrasound. T = 50 °C.

**Figure 7 polymers-15-01968-f007:**
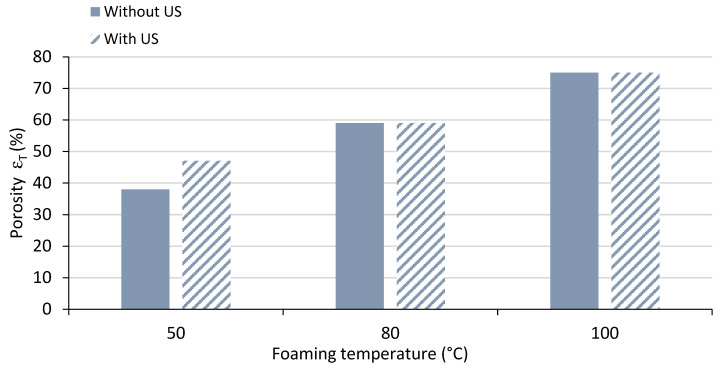
Porosity of foamed PMMA without or with US, in a conventional US bath.

**Figure 8 polymers-15-01968-f008:**
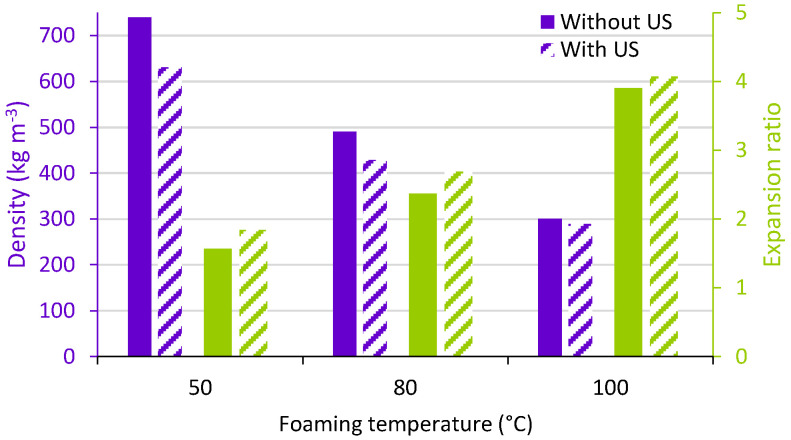
Density and expansion ratio of foamed PMMA in a conventional US bath. (solid bars: without US, striped bars: with US).

**Figure 9 polymers-15-01968-f009:**
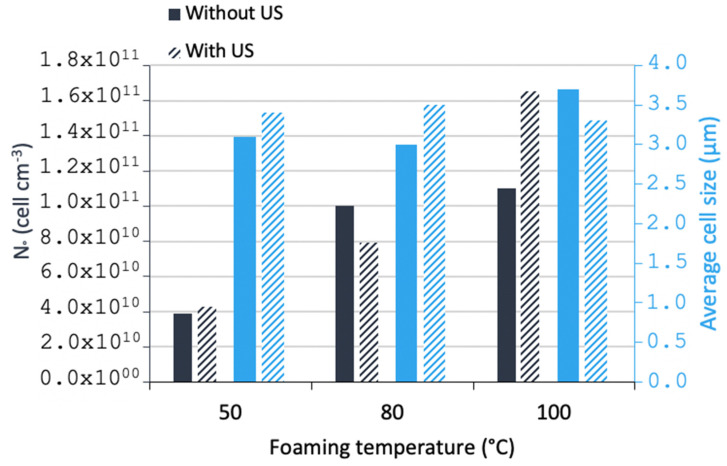
Cellular density and average cell size of foamed PMMA in a conventional US batch.

**Figure 10 polymers-15-01968-f010:**
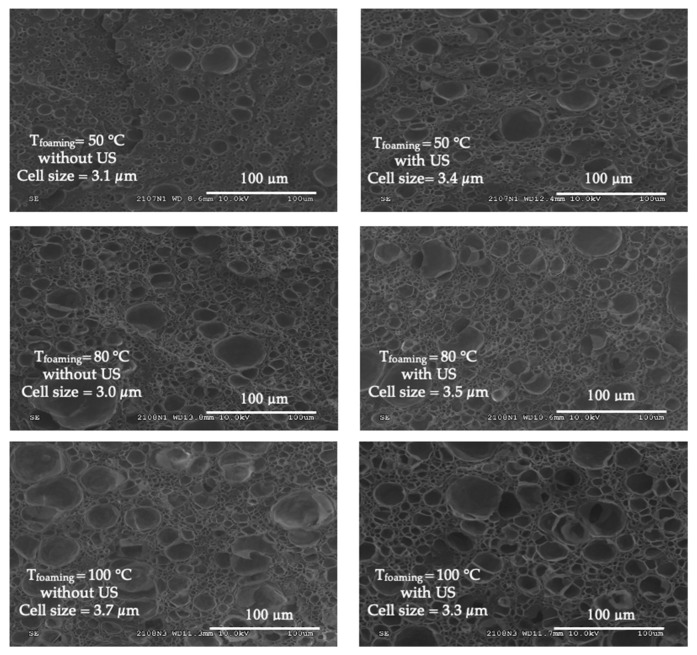
SEM morphologies of foamed PMMA at different temperatures without (**left**) and with conventional US (**right**).

**Figure 11 polymers-15-01968-f011:**
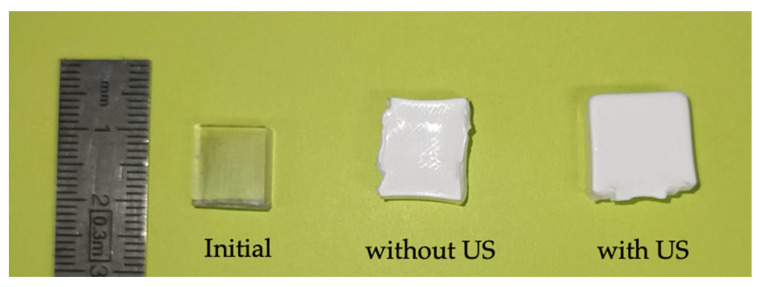
PMMA samples foamed by sc-CO_2_-assisted batch foaming with and without localised US. T_foaming_ = 80 °C.

**Figure 12 polymers-15-01968-f012:**
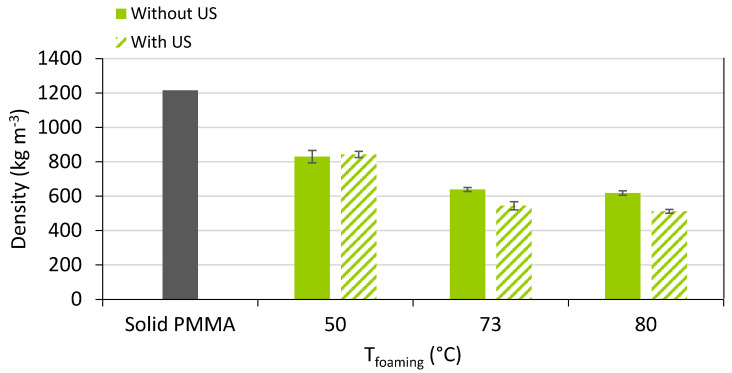
Density of PMMA foams produced by sc-CO_2_ assisted batch foaming with and without localised US.

**Figure 13 polymers-15-01968-f013:**
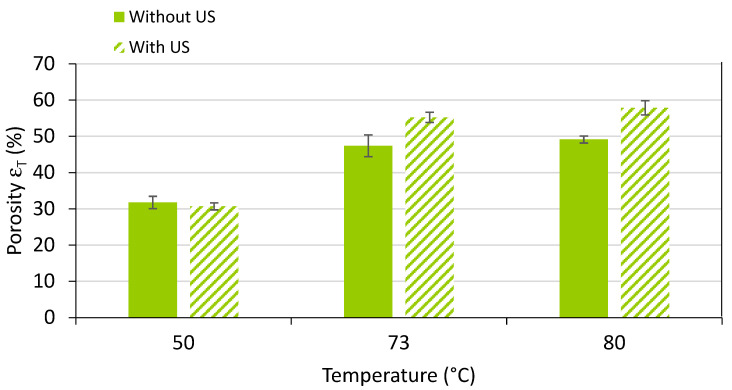
Porosity of solid PMMA and its foams produced by sc-CO_2_-assisted batch foaming without and with localised US.

**Figure 14 polymers-15-01968-f014:**
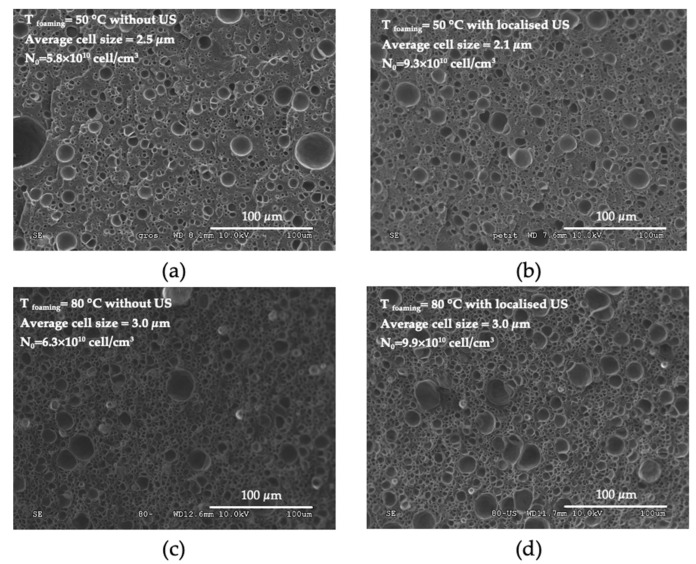
SEM microphotographs of PMMA foams produced by sc-CO_2_-assisted batch foaming without (left) and with (right) localised US. (**a**,**b**) T = 50 °C, (**c**,**d**) T = 80 °C.

**Figure 15 polymers-15-01968-f015:**
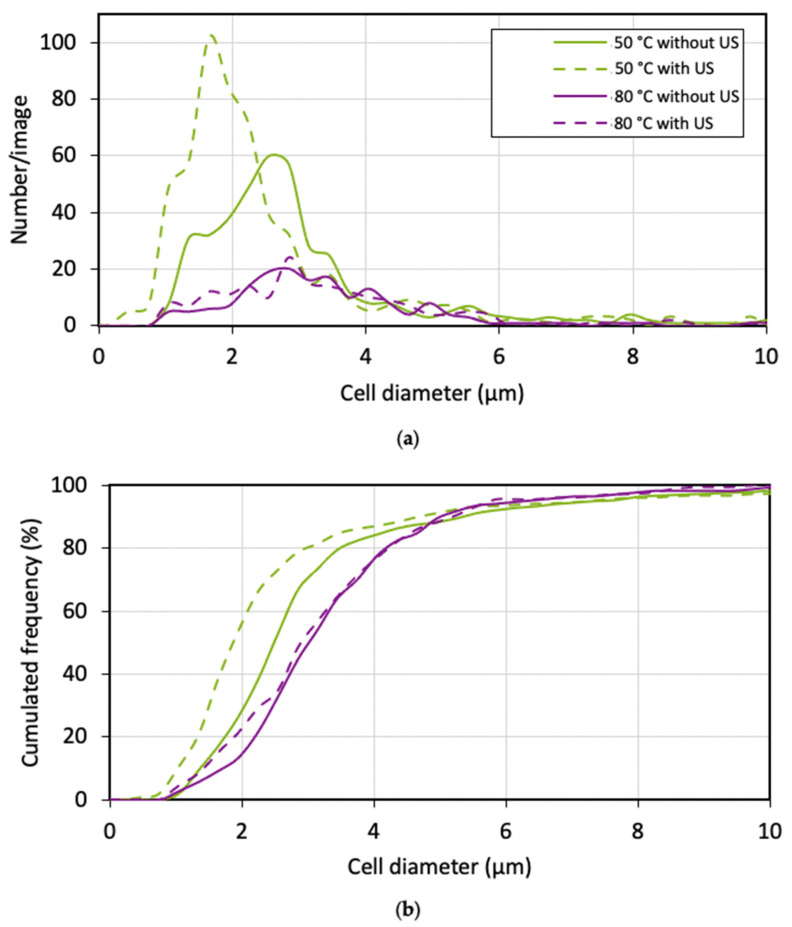
(**a**) Number and (**b**) cumulated frequency cell size distributions of PMMA foams produced by sc-CO_2_-assisted batch foaming without and with localised US.

**Figure 16 polymers-15-01968-f016:**
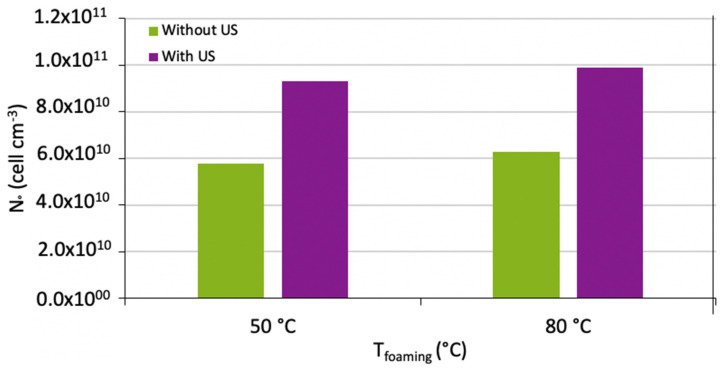
Cell density of PMMA foams produced by sc-CO_2_-assisted batch foaming without and with localised US.

**Figure 17 polymers-15-01968-f017:**
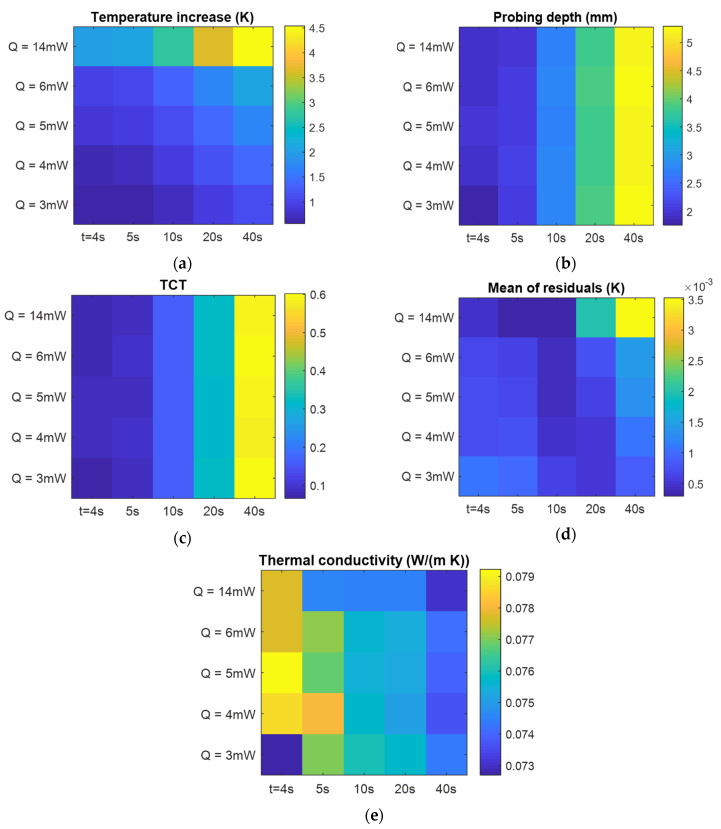
Evaluated preliminary criteria for thermal plane source characterization according to heating power and measurement time: (**a**) total temperature increase; (**b**) estimated probing depth; (**c**) ratio of measurement time to characteristic time; (**d**) mean of residuals; (**e**) the estimated thermal conductivity.

**Figure 18 polymers-15-01968-f018:**
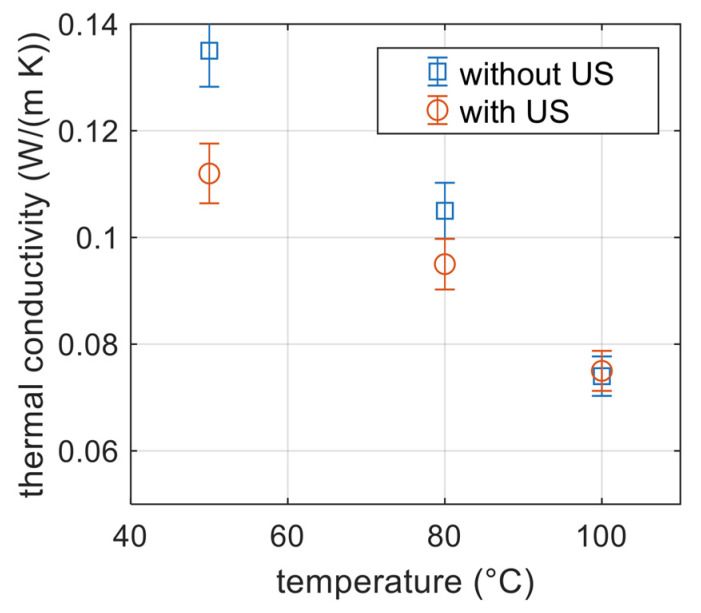
Thermal conductivity of foamed PMMA obtained at different temperatures without and with conventional US.

## Data Availability

The data that support the findings of this study are available on request from the corresponding author.

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
