# Peer review of "Polymer Supercritical CO_2_ Foaming under Peculiar Conditions: Laser and Ultrasound Implementation"

_polymers, 2023, doi:10.3390/polym15081968_

Round 1
Reviewer 1 Report
This work studies solid-state two-step CO2-foaming of PMMA. The results are interesting, while revisions are needed:
1. The Abstract is not well written and should be refined. Please provide a brief but concise summary of background, methods, results and discussions.
2. There are a number of grammatical errors or spelling errors which must be corrected. For example, caption of Figure 2, “Depressurisation profil” should be “Depressurisation profile”. The writing should also be refined to conform to academic writing standard.
3. In Introduction, the author reviewed a large number of existing works for polymer foaming with ultrasound. However, what is the issue with these existing works? What is the novelty and significance of the authors’ work? Please point out.
4. How are the parameters for laser assisted foaming and ultrasonic assisted foaming determined? Are there any preliminary tests?
5. It is suggested to re-draw the scale bar in SEM images.
6. In Figure 8, legend for “With US” is missing.
Author Response
please find the attachment with our answers to the reviewers

Reviewer 2 Report
The manuscript “Polymer supercritical CO2 foaming under peculiar conditions: Laser and UltraSounds implementation” by J.A.V. Jiménez et al. is devoted to the pilot studies of the two-step scCO2-foaming of PMMA or PMMA-block copolymer blends using the additional action of laser radiation or ultrasound. The discussed techniques and obtained results could be of interest to the community of the “Polymers” readers specializing in the synthesis of functional materials based on foamed polymers. However, reading the manuscript raises certain questions and comments that should be taken into account before making a final decision on the possibility of acceptance.
1. The style of presentation is sometimes hard to follow. I recommend a substantial revision of the Section “Materials and Methods”; It is better to start with a description of the first stage (saturation of the raw polymers with carbon dioxide), paying more attention to the physicochemical aspects of this process. On what basis the parameters of the "saturation/depressurization" stage were chosen? These parameters are: 24-hour exposition in the scCO2 atmosphere at 40 degrees and 10 MPa, rapid release of pressure from 10 to 7 MPa followed by slower release during approximately 60 s. How changes in these parameters affect the result? Are the authors sure that “the depressurisation profile shown in Figure 2” (line 155) provides the best result. In my opinion, such kind of depressurization can cause formation of an ensemble of nuclei in the processed polymer, which are the germs for future pores at the next stage. 2. The used materials should be described in more details (are they crystalline or amorphous; if amorphous, what is the temperature range for glass transition and how is it affected by CO2 impregnation, etc.). The knowledge of these points is important for better choice of foaming parameters, and reference like "Material details, characterization and foaming methods are given in another paper [5] (line 124)" are insufficient.
3. Please, insert multiplication sign instead of decimal separator in the numerator in the first parenthesis of Eq. 2.
4. The histograms of pore distributions are desirable.
5. The physical mechanisms of laser and ultrasound action should be discussed in more detains. It is not clear why do the authors use CO2 laser with radiation absorbed in shallow layers of a foamed polymer? Application of lasers with shorter wavelengths (e.g., Nd:YAG laser at high repetition rates) can provide bulk absorption and, accordingly, better heating conditions.
6. I am surprised that “But the real temperature of material during foaming is unknown, depends on a lot of factors (laser power, scan speed, distance to surface) … (lines 410-411). Estimation of the laser heating temperature is a rather traditional problem in laser technologies and the corresponding methods are known. It is necessary to know only the absorption coefficient of a material, which can be found in literature and databases.
Author Response

(The authors gave the same response as above.)

Reviewer 3 Report
The manuscript describes the study of ultra-sound and laser-assisted foaming of PMMA. The authors study morphology, porosity, cellular density and thermal conductivities of the obtained PMMA samples and compare the values. It is a thorough experimental work that might be of interest for the readers of the Polymers journal. However, before publishing, a few important issues need to be sorted out. Please see my comments below.
1. Experimental
Experimental procedure for preparation of the samples is not clear. Were the samples treated with sc-CO2 first and then with laser and US, or was it the other way around? For the sc-CO2 procedure, authors use different units in 2.1 and 2.2.2, while in 2.2.1. the sc-CO2 procedure is not described at all. The authors should carefully address these issues in the revised manuscript.
2. Experimental, Results and discussion.
The authors compare LS treatment and US treatment with the samples marked as “Without”. Yet, there is no mention of how those samples are prepared in the experimental section.
3. Experimental, Results and discussion
In figures 7, 8, 9, 12, 13 the authors compare the properties of the foamed PMMA for the samples ‘with US’ and ‘without’. We already mentioned that the ‘without’ samples are not described in the experimental section. Further, the data is for different temperatures and it is not clear, are those the temperatures of co2 treatment or US treatment? If it is the first, then why only 40oC temperature for the co2 treatment is mentioned in the experimental section? If it is the second, then how are the without samples obtained? I suggest that the authors provide a table where for all sample types the procedure is given, with detailed parameters, such as temperature, pressure, treatment time, etc.
4. Conclusions
Lines 405-406:
The process is a two-step solid-state batch scCO2 process; foaming occurs in the second heating step with the aid or assistance of LS or US.
It is unclear which step is the first and which is the second. If the sc CO2 process is the first, then this statement in conclusions is incorrect, as even the results of this work show (the porosities of the samples “without” are growing, the densities and thermal conductivities are decreasing, indicating that the samples were foamed. )
5. Minor suggestion:
Lines 39-40, 77, 116.
The abbreviations are used inconsistently – some abbreviations are given without full names (like PMMA, PET, PVC, HMSPP), in one case the abbreviation precedes the full name (line 116). Please carefully unify the abbreviations.
Author Response

(The authors gave the same response as above.)

Round 2
Reviewer 1 Report
The authors have properly addressed the review comments and it can be accepted for publication now.
Reviewer 2 Report
The provided revisions and additional comments have made the manuscript more suitable for publication. In the current form, it can be accepted.